# Oligomeric Architecture of Mouse Activating Nkrp1 Receptors on Living Cells

**DOI:** 10.3390/ijms20081884

**Published:** 2019-04-16

**Authors:** Ljubina Adámková, Zuzana Kvíčalová, Daniel Rozbeský, Zdeněk Kukačka, David Adámek, Marek Cebecauer, Petr Novák

**Affiliations:** 1Laboratory of Structural Biology and Cell Signaling, Institute of Microbiology, The Czech Academy of Sciences, Vídeňská 1083, 14220 Prague 4, Czech Republic; ljubina.adamkova@biomed.cas.cz (L.A.); rozbesky@gmail.com (D.R.); zdenek.kukacka@biomed.cas.cz (Z.K.); adamek.david@outlook.com (D.A.); 2Department of Biophysical Chemistry, J. Heyrovsky Institute of Physical Chemistry, The Czech Academy of Sciences, Dolejškova 2155/3, 18223 Prague 8, Czech Republic; zuzu.qmail@gmail.com (Z.K.); marek.cebecauer@jh-inst.cas.cz (M.C.); 3Department of Biochemistry, Charles University, Hlavova 8, 12843 Prague 2, Czech Republic

**Keywords:** Nkrp1, dimerization, Förster resonance energy transfer, disulfide bond arrangement, cysteine

## Abstract

Mouse activating Nkrp1 proteins are commonly described as type II transmembrane receptors with disulfide-linked homodimeric structure. Their function and the manner in which Nkrp1 proteins of mouse strain (C57BL/6) oligomerize are still poorly understood. To assess the oligomerization state of Nkrp1 proteins, mouse activating EGFP-Nkrp1s were expressed in mammalian lymphoid cells and their oligomerization evaluated by Förster resonance energy transfer (FRET). Alternatively, Nkrp1s oligomers were detected by Western blotting to specify the ratio between monomeric and dimeric forms. We also performed structural characterization of recombinant ectodomains of activating Nkrp1 receptors. Nkrp1 isoforms c1, c2 and f were expressed prevalently as homodimers, whereas the Nkrp1a displays larger proportion of monomers on the cell surface. Cysteine-to-serine mutants revealed the importance of all stalk cysteines for protein dimerization in living cells with a major influence of cysteine at position 74 in two Nkrp1 protein isoforms. Our results represent a new insight into the oligomerization of Nkrp1 receptors on lymphoid cells, which will help to determine their function.

## 1. Introduction

Natural killer (NK) cells are innate lymphocytes that play an essential role against virally infected and tumor cells [1]. The specificity of NK cells is controlled by a repertoire of activating and inhibitory cell surface receptors [2], which are divided into two distinct structural groups: the immunoglobulin superfamily and the C-Type Lectin-like Receptor (CTLR) family [3]. Mouse Nkrp1 proteins belong to the CTLR family and have been initially compared with their rat homolog. The rat Nkrp1 is a type II transmembrane disulfide-linked homodimer [4] which stimulates NK cells [5,6], especially phosphatidylinositol turnover, Ca2+ mobilization [7], cytolytic processes [6] and cell degranulation [6]. These findings have been subsequently extrapolated to the mouse Nkrp1 molecule. The murine Nkrp1 gene family encodes five members with either an activating or an inhibitory function; isoforms Nkrp1a, c and f are activating, and b/d and g have inhibitory function [4,8,9,10,11,12]. The first receptor identified to be selectively expressed by NK cells was the mouse NK1.1 antigen [13], later named Nkrp1c. The NK1.1 was defined as a 39 kDa glycoprotein [14] capable of inducing lytic functions of NK cells after cross-linking with a specific monoclonal antibody (PK136) [15]. Characterization of other NK alloantigens [16] and molecular cloning of mouse Nkrp1 proteins [8,17] provided a new insight into the nature of this receptor. In 1992, Ryan et al. [17] demonstrated that mouse NK1.1 antigen exhibits a homodimeric state when immunoprecipitated by the PK136 antibody, similar to its rat homolog.

An increasing body of evidence in recent years has provided support for Nkrp1 binding of C-type lectin-related ligands (Clr) [18,19], which are genetically linked to the Nkrp1 genes [18,20]. Recent work has shown that a structurally divergent, viral Ig-like m12 immunoevasin is also a ligand for the inhibitory Nkrp1b and activating Nkrp1a/c receptors [21].

In previous studies, all Nkrp1 receptors were considered to be disulfide-linked dimers. However, the number of extracellular cysteine residues varies, e.g., in between activating Nkrp1 proteins expressed in C57BL/6 mouse strain. Four cysteines of the extracellular domain are highly conserved and form two intramolecular disulfide bridges [22]. Intriguingly, besides simply connecting the CTLD to the plasma membrane, the stalk region might be a key player in the oligomeric architecture of the Nkrp1 receptors. Experimental data support the existence of dimers for Nkrp1b/d [18] and Nkrp1c proteins [17]. The oligomeric state of the remaining murine isoforms is unknown. Furthermore, no live cell imaging data for any of the Nkrp1 receptors have been published so far.

To investigate the oligomerization state of murine activating Nkrp1 proteins, a panel of native receptors and their stalk cysteine-to-serine mutants fused to EGFP protein was expressed in mammalian cells. Using microscopic techniques, the protein subcellular localization was studied. A monomer/dimer distribution on the plasma membrane of living cells were monitored by a variant of Förster resonance energy transfer (FRET) technique—fluorescence anisotropy (also called homo-FRET). The existence of monomer and dimer isoforms of mouse Nkrp1 was further verified by immunoblotting after solubilization of cell membranes. Additionally, ectodomains of native activating Nkrp1 isoforms (a, c1, c2, and f) were heterologously produced, refolded and structurally characterized. All our data indicate tight control of the dimeric state of Nkrp1 isoforms by specific cysteine residues present in their stalk domains.

## 2. Results

### 2.1. Surface Localization of Mouse Activating Nkrp1 Proteins in Mammalian Cells

To function as NK receptors, Nkrp1 proteins must localize to the cell surface. We have thus evaluated the surface expression of EGFP-fusion of murine activating Nkrp1 proteins in cultured human Jurkat cells which do not express murine receptors endogenously. Endogenous expression of Nkrp1 proteins would affect FRET analysis used to determine dimerization state of the receptors (see below). As shown in Figure 1A–D, all tested Nkrp1 isoforms were efficiently sorted to the plasma membrane of Jurkat T cells.

Cysteines in the extracellular part are often involved in dimerization of surface receptors [23]. In the stalk region, there are two cysteines in Nkrp1a, three in Nkrp1c1, two in Nkrp1c2 and sole a cysteine residue in Nkrp1f which can potentially form intermolecular disulfide bonds. Cysteine-to-serine mutants were prepared to determine the involvement of these residues in dimerization of murine activating Nkrp1 proteins. We were thus interested in whether these cysteine-to-serine mutations influence protein expression and cell surface localization of the receptors. None of these mutations altered distribution of tested variants (Figure A1). Cysteine-to-serine mutants were localized at the surface of Jurkat cells similar to their native variants. All receptor variants also exhibited localization to the Golgi apparatus which was probably caused by transient expression of these proteins. It is worth mention that cysteine-to-serine mutation instead cysteine-to-alanine was selected because serine and cysteine differ only in the swap of a sulfur atom with oxygen and share a high level of similarity in terms of size and structure. Furthermore, all mutations were located only in the stalk region which is expected to be fully exposed to the solvent. Thus, the substitution to serine residue, rather than alanine residue, which is more hydrophobic, appears to be more meaningful.

### 2.2. Stoichiometry of Nkrp1 Proteins in Mammalian Cells

#### 2.2.1. Homo-FRET Analysis of Nkrp1 Proteins in Living Jurkat Cells

A sufficient expression and surface localization of all Nkrp1 variants enabled direct analysis of their oligomerization state in living cells by fluorescence anisotropy (donor-donor energy transfer). This technique was preferred for studies of Nkrp1 homo-oligomerization for two main reasons: First, results of more common hetero-FRET analysis (energy transfer from one fluorescent dye–donor–to another dye–acceptor) are strongly affected by expression levels of donor and acceptor. The use of a single fluorescently labelled protein (as for the anisotropy technique) eliminates this limitation. Second, stochastic pairing of proteins tagged with the same fluorophore (e.g., GFP-GFP) reduces sensitivity of hetero-FRET measurements to homo-dimers. Since FRET measurements using fluorescent proteins in living cells are prone to low sensitivity [24], it is important to select conditions which maximize sensitivity. This is fulfilled by measuring homo-FRET of exogenous fluorescently labelled proteins in the absence of endogenous interacting molecules. On the other hand, homo-FRET measurements are strictly semi-quantitative [25] and require calibration by monomeric and dimeric controls. Till this point, we used monomeric LAT (linker of activated T cells) and dimeric CD8α proteins as controls for homo-FRET oligomerization analysis of Nkrp1 proteins. As anticipated, steady-state anisotropy of LAT was significantly higher than that of CD8α, corresponding to CD8α forming dimers or higher oligomeric states and monomeric state of LAT (Figure 2A,B).

Figure 2A also presents steady-state anisotropy values determined for Nkrp1 proteins transiently expressed in Jurkat cells in the form of EGFP fusion proteins. Almost identical values of steady-state anisotropy were measured for dimeric control (CD8α) and Nkrp1c1. These data from living cells thus support the earlier observation that murine Nkrp1c1 forms stable dimers [17]. Isoforms c1 and c2 differ only by three amino acid residues in the stalk region, one of which is cysteine at the position 88 (with respect to Nkrp1c1 numbering; Figure A2). Since Nkrp1c1 has one additional cysteine compared to the c2 isoform, it is important to determine its involvement in protein dimerization. The data obtained reveals comparable levels of dimerization for Nkrp1c1, Nkrp1c2, as well as Nkrp1f isoforms (Figure 2A). Cysteine 88, which is also missing in the stalk region of Nkrp1f, therefore cannot be involved in dimerization of Nkrp1 receptors. Much higher steady-state anisotropy was measured for Nkrp1a1 which contains cysteine 88 in its stalk region but lacks one of the cysteine residues at the positions 74–75 when compared to the native Nkrp1c1 isoform. This result suggests less dimeric character of this isoform. However, Nkrp1a1 anisotropy also diverges from a monomeric control (LAT; Figure 2A), indicating the existence of some proportion of Nkrp1a1 dimers/oligomers which are in equilibrium with monomers on the surface of living Jurkat cells. Altogether, the steady-state anisotropy measurements imply the existence of diverse populations of Nkrp1 receptors in the plasma membrane of living Jurkat cells (Figure 2A).

Cysteine-to-serine stalk mutants of Nkrp1 proteins support these conclusions (Figure 2B). Whereas no change in measured steady-state anisotropy was observed for single residue mutants (C74S or C88S) of Nkrp1a1, its dimerization was significantly reduced when both cysteines were changed for serines (C74,88S variant; Figure 2B). On the contrary, a mutation of single cysteine (C74S) strongly reduced dimerization level of Nkrp1f. Indeed, steady-state anisotropy values measured for this mutant corresponded to that of double cysteine mutant (C74,88S) of Nkrp1a1, (Table A1) indicating that this mutation leads to similar changes in the organization of Nkrp1 on the surface of Jurkat cells. Steady-state anisotropy values measured for Nkrp1c1_C74,75S double mutant correspond to that of a single cysteine mutant C74S of Nkrp1a. Cysteine residue at the position 88 remains intact in these mutants. These live-cell data thus demonstrate formation of tight dimers by c1, c2 and f isoforms but much looser assemblies of the a1 isoform at the plasma membrane of Jurkat cells. The formation is regulated by exact positions of cysteine residues in the stalk region of the Nkrp1 proteins. The overview of cysteines mutation represents Figure 2C.

#### 2.2.2. Western Blotting of Nkrp1 Protein Variants in COS-7 cell Lysates

To further support diverse oligomerization states of Nkrp1 isoforms in mammalian cells, we determined the molecular weight of Nkrp1 protein complexes by immunoblotting of cell lysates after non-reducing acrylamide gel electrophoresis. This biochemical approach required a larger number of transfected cells for each sample, thus we employed COS-7 cells for a larger scale protein expression. In addition, the use of such a different cell type (Rhesus monkey kidney fibroblasts) provides another model to investigate cell-type independent oligomerization of Nkrp1 receptors. COS-7 cells were transiently transfected with native isoforms and cysteine-to-serine mutants of EGFP-Nkrp1 fusion proteins. Cells were lysed on ice in the presence of iodoacetamide to block free thiol groups of cysteines and prevent non-specific disulfide bond scrambling. Cell lysates were then analyzed by immunoblotting with anti-GFP antibody. Under reducing conditions, all EGFP-Nkrp1 proteins migrated as monomers (Figure 3A). Results from non-reducing conditions showed a presence of a mixture of monomers, dimers and higher order oligomers for almost all tested EGFP-Nkrp1 proteins including cysteine-to-serine stalk mutants (Figure 3B). In agreement with FRET results, dimers were the dominant oligomeric state detected for isoforms c1, c2 and f. In these experiments, a larger proportion of monomers was found for Nkrp1c2 compared to Nkrp1c1 proteins. A lower expression level of Nkrp1a prevents its direct comparison to the abovementioned dimeric isoforms but the presence of both, the protein monomers and dimers is well detectable, as is for its mutants (Figure 3B). Interestingly, only dimers were detected for C88S single residue mutant of Nkrp1a1. This result contrasts with the anisotropy values measured in living Jurkat cells, which are comparable to the native Nkrp1a protein, and may indicate slight differences between the cell environment or limitations of this method. Decreased appearance in the level of dimers found for C74,88S double mutant compared to its native variant supports lower oligomerization of this mutant observed by homo-FRET (Figure 2B). A similar effect was observed for C74S mutant of Nkrp1f. On the contrary, only minor increase of monomers was observed for C74,75S double mutant of c1 isoform which, similar to the native protein, exhibited almost exclusively dimeric character in these experiments (Figure 3B). These biochemical results indicate that Nkrp1 receptors form dimers with isoform-specific efficiency and, thus, support the homo-FRET data acquired in living lymphoid cells. Actin was used as a loading control (Figure 3C).

### 2.3. Refolding, Purification and Evaluation of Disulfide Bonds in Recombinant Murine Activating Nkrp1 Proteins

To better understand how extracellular cysteines of Nkrp1 receptors are involved in the formation of intra- and inter-molecular disulfide bridges, recombinant ectodomains of these proteins (Nkrp1^ECTO^) were expressed in E. coli, purified, refolded and structurally characterized (see Material and Methods for more details). The final yields of purified Nkrp1 ectodomains per liter of bacterial culture were: 12 mg of Nkrp1a^ECTO^, 1-2 mg of dimeric Nkrp1c1^ECTO^, 8 mg of Nkrp1c2^ECTO^, and 6.5 mg of Nkrp1f^ECTO^. Size-exclusion chromatography of purified protein ectodomains was performed to characterize the homogeneity of the samples (Figure A3). Receptor isoforms a1 and c2 show almost exclusively a single peak in the chromatograms indicating their high purity and low variation in stoichiometry. On the contrary, a broader spectrum of peaks was detected on chromatograms for ectodomains of isoforms c1 and f (Figure A3). This suggests the coexistence of two or more oligomeric states under these conditions. Main peaks were used to isolate individual forms of recombinant Nkrp1 ectodomains and analyzed by SDS-PAGE in a 15% polyacrylamide gel under reducing and non-reducing conditions (Figure 4). Ectodomains of isoforms c1 and f were found to form both monomers and dimers. Only monomeric forms were found for the remaining isoforms (a1 and c2).

To further demonstrate the capacity of Nkrp1 ectodomains to dimerize and the oxidative state of cysteines, the exact mass of purified recombinant Nkrp1^ECTO^ proteins was determined by electrospray ESI-FT-ICR mass spectrometry. Monoisotopic masses of Nkrp1^ECTO^ proteins in native and reducing conditions are summarized in Table 1. The difference between monoisotopic mass of oxidized and reduced proteins corresponded to the predicted monoisotopic mass for proteins with all paired cysteines. Ectodomains of Nkrp1a and Nkrp1c2 possess four disulfide bonds, Nkrp1c1^ECTO^ protein has four disulfide bridges and one free cysteine, and Nkrp1f^ECTO^ contains three cysteine pairs and one unpaired cysteine residue. Moreover, the Nkrpc1^ECTO^ and Nkrp1f^ECTO^ bearing odd number of cysteine residues were found in both monomerric and dimeric states.

The ectodomains of mouse Nkrp1 proteins contain six to nine cysteine residues. We were thus interested to know which cysteines form intra- and inter-molecular bonds. Purified recombinant Nkrp1^ECTO^ proteins were subjected to proteolytic reactions by trypsin, AspN and GluC proteases according to published protocol [26] and disulfide bond mapping was performed. Peptides generated by the enzymatic digestion containing disulfide-linkages are summarized in Table A4. Standard cysteine linkages of C-Type Lectin-like Domain (CTLD) were observed for all activating Nkrp1 proteins (Figure 5, grey area). The only differences were found in the stalk region. Intramolecular bonds between C74-C88 were observed within Nkrp1a^ECTO^ and C74-C75 within Nkrp1c1^ECTO^ and Nkrp1c2^ECTO^ molecules. Moreover, intermolecular connections between C88-C88 and C74-C74 were identified within Nkrp1c1^ECTO^ and Nkrp1f^ECTO^ molecules, respectively. Such observation was supported by the fact that C88 of Nkrp1c1^ECTO^ and C74 of Nkrp1f^ECTO^ were identified as an unpaired cysteine bearing cysteamine moiety when the monomeric forms were analyzed. Schematic illustration of the final disulfide arrangement of recombinant Nkrp1 ectodomains is shown in Figure 5. Structural models of Nkrp1a^ECTO^ and Nkrp1c1^ECTO^ proteins (Figure A4) display positions of the surface available to cysteine residues potentially available for intermolecular disulfide bonding.

## 3. Discussion

Dimerization and oligomerization can positively or negatively regulate function of proteins without a need for their de novo synthesis or downregulation. In biological systems, many key cell-surface proteins such as G-protein-coupled receptors [27,28], ion channels [29], insulin receptor [30], receptor tyrosine kinases [31], members of the cytokine receptor superfamily [32,33] and enzyme complexes [34,35,36], can exist as homo- or hetero-di/oligomers. Historically, functional and biochemical data acquired in the experiments using rat Nkrp1 protein were widely and directly extrapolated to Nkrp1 molecules from other species (including mouse). Virtually all Nkrp1 receptors have been previously schematically depicted as disulfide-linked homodimers. However, to date, there is literally just a single study demonstrating homodimerization of mouse Nkrp1c upon antibody crosslinking [17]. To approach for the function of mouse Nkrp1 proteins, which is still unclear, understanding of basic protein structure is fundamental and obligatory.

Here, we determined the oligomeric state of mouse activating Nkrp1 proteins in living mammalian cells. As expected, a good surface expression was found for all tested Nkrp1 proteins in several cell lines (Jurkat T-cells, COS-7, RAW264.7 and HeLa cells; Figure 1, A1 and ref. [37]). Interestingly, mouse Nkrp1 receptors conjugated to EGFP protein were toxic for the mouse macrophage RAW264.7 cells [37]. On the contrary, good protein expression was achieved in Jurkat (Figure 1), COS-7 (Figure 3) and HeLa cells [37]. Cysteine-to-serine mutations in the stalk region did not significantly affect protein localization in Jurkat cells (Figure A1). For live cell imaging and steady-state anisotropy measurements, suspension Jurkat cell line was utilized considering easier manipulation during experiments and their lymphoid origin. Moreover, their round cell morphology is advantageous for homo-FRET data analysis. Monkey kidney cells, COS-7, were used to investigate oligomeric state of Nkrp1 receptors by biochemical approach due to higher yields of proteins required for this technique.

Our results from fluorescence microscopy (Figure 2), biochemical (Figure 3) and structural analyses (Figure 4 and Figure 5) point towards complex regulation of oligomerization of murine Nkrp1 receptors. Native Nkrp1c1, Nkrpc2 and Nkrp1f receptors form prevalently dimers (or higher order structures) on the surface of living lymphoid cells (Figure 2A). The native form of Nkrp1a displayed a more monomeric character when compared to the other isoforms (Figure 2A). Steady-state anisotropy values measured for Nkrp1a differed significantly from both controls, monomeric LAT and dimeric CD8α. Moreover, higher abundance of monomers was detected in lysates of cells expressing Nkrp1a than those expressing isoforms c1 or c2. These cell-based data indicate that Nkrp1a can form dimers, but these are less stable than highly abundant and stable dimers of isoforms c1, c2 and f. These data are also supported by purified recombinant proteins. The Nkrp1a^ECTO^ protein was prepared as a monomer (Figure 4), which is in accordance with already published results [38,39]. However, it was previously demonstrated that the Nkrp1a can form non-covalent dimers through its stalk region [40].

Mapping of intermolecular disulfide bridges responsible for receptor dimerization was rather straightforward for Nkrp1f which contains a sole cysteine residue in its stalk region. Strong prevalence of dimeric population on the plasma membrane was found using homo-FRET analysis (Figure 2A). Since mass spectrometric analysis of recombinant Nkrp1f^ECTO^ protein determined cysteines of CTLD forming intramolecular connections and C74 as a dimerization interphase (Figure 5 and Table A4), mutant variant lacking C74 was created to confirm such observation in living cells. Indeed, mainly monomers of Nkrp1f_C74S mutant were found on the plasma membrane using homo-FRET and immunoblotting analysis (Figure 2B and Figure 3).

The remaining isoforms Nkrp1a, Nkrp1c1 and Nkrp1c2 share the same arrangement of cysteine bridges in CTLD of the receptor with Nkrp1f (Figure 5 and Table A4). However, they contain two or three cysteine residues in the stalk region. Thus, more cysteine-to-serine mutants had to be designed to study the effect of cysteine residues on their oligomerization. In the case of Nkrp1a isoform, point mutations of C74S or C88S showed no significant effect on protein oligomerization in living Jurkat T cells. However, similar to Nkrp1f_C74S mutant, prevalently monomeric state was observed for a double Nkrp1a_C74,88S mutant (Figure 2B and Figure 3). These two mutants lack cysteine residue in their stalk region and indicate a comparable ratio of Nkrp1 dimers (or oligomers) which are in equilibrium with monomers on the plasma membrane of Jurkat cells. Altogether, our data from homo-FRET and immunoblotting suggest that all stalk cysteines play a role in Nkrp1 dimerization. It is likely that a small portion of non-covalent dimers are presented on the cell membrane. This assumption reflects the Nkrp1f state with the respect to their sequence similarity and it is further supported by dominant covalent dimer formation on western blot (Figure 3B). In the case of Nkrp1a receptor, a non-covalent association through the stalk region, which was also previously demonstrated by Kolenko et al. [40], influences a full conversion into the monomer for the double mutant as well (Figure 2B and Figure 3B). Furthermore, it has been shown that a related human activating receptor NKp65 can form non-disulfide-linked homodimers [41] but interacts with its non-covalent dimeric ligand (keratinocyte-associated C-type lectin, KACL) as a monomer [42].

Nkrp1a lacks free cysteine residue in its stalk region. It was, therefore, expected that single cysteine mutant will have limited impact on dimerization of this receptor. Interestingly, dual mutation of cysteine residues (C74,88S) significantly reduced dimer levels of Nkrp1a on the cell surface. We speculate that such dual mutation affects the overall structure of the ectodomain, thus destabilizing non-disulfide dimers of Nkrp1a. Homo-FRET and immunoblotting results suggest that the Nkrp1c1 receptor is expressed predominantly as a dimer on the plasma membrane (Figure 2A and Figure 3B). The natural deletion in the protein sequence of the Nkrp1c2 isoform, which includes cysteine residue corresponding to C88 of the c1 isoform, is also present on the cell surface predominantly as a dimer (Figure 3B). Pursuant to the disulfide bond determination by mass spectrometry (Figure 5), the free stalk cysteine C88 was considered to serve for covalent dimerization of Nkrp1c1. However, natural variant lacking this residue, Nkrp1c2, was found to form high levels of dimers at the surface of living cells what contradicts this thesis. We thus speculate that other stalk residues, specifically C74 or C75, regulates dimerization of Nkrp1c1 and Nkrp1c2. Indeed, our homo-FRET data demonstrate strongly reduced capacity of Nkrp1c1_C74,75S double mutant to form oligomers on the cell surface (Figure 2B). Therefore, cysteine required for Nkrp1c1 oligomerization in live cells is not the unpaired C88 as predicted from dimeric Nkrp1c1^ECTO^ protein refolded in vitro, but residues C74 or C75. From a steric point of view, the disulfide bond between the two adjacent cysteines should be unfavorable. In nature, they are very rare and unique. However, vicinal S-S bonds introducing a strain into a protein structure have been documented [43,44,45].

Our study of Nkrp1 oligomerization displays that all stalk cysteines are more or less relevant in protein dimerization. Similar behavior was observed for oligomerization of a C5a receptor, where C144 displayed the fastest kinetics of dimer formation in comparison to other cysteine residues in the protein sequence. [46]. One can also consider similarity of Nkrp1 receptors to the Ly49 proteins, another mouse C-Type Lectin-like Receptor family. The stalk of Ly49 receptors enables protein dimerization and two distinct receptor conformations [47]. However, despite being structurally and functionally similar to Ly49 receptors of mouse, the stalk region of Nkrp1a^ECTO^ does not alter the ligand binding domain [39] as it has been published for the Ly49 receptors [47,48]. Most likely, the shorter stalk of Nkrp1a protein does not favor cis/trans reorientation of the extracellular C-Type Lectin-like Domain (CTLD).

In summary, mouse activating Nkrp1 proteins are expressed on the cell surface as dimers with diverse levels of coexisting monomers. The level of surface dimerization seems to be regulated by the precise positioning of cysteine residues in the stalk region and not exclusively by the presence of putative unpaired cysteines. Coexistence of monomeric and dimeric proteins on the cell surface has been documented for other essential membrane proteins. In the case of CAPRI protein (a member of GTPase-activating proteins), the oligomerization process is controlled by calcium and activates different cellular signaling pathways based on its oligomerization state [49]. In case of Nkrp1 proteins, the function and purpose of such balance between monomers and dimers in the plasma membrane is still unknown. It is believed that signaling through Nkrp1 molecules involves receptor dimers [50]. By contrast, constitutively active covalent Nkrp1 dimers on the cell surface are unlikely from the immunological point of view. It is, therefore, tempting to speculate that the coexistence of monomers and dimers plays a regulatory role for the function of these receptors. The most recent crystal structure of Nkrp1b:m12 complex has revealed the engagement of monomeric Nkrp1b enclosed by the m12 monomer [21], as opposed to previously suggested interaction of a receptor as a dimer [4,5,18]. Contrary to cytomegalovirus-encoded protein m12, the interaction of homodimeric host-encoded ligand Clr-b with Nkrp1b receptor showed that additional avidity of non-classic Nkrp1b dimer is needed [51]. This suggests an avidity driven manner of ligand binding. On the other hand, function of Nkrp1b is to inhibit NK cell activation, not to activate.

Recently, it has been reported that mouse Nkrp1a and Nkrp1c also bind to a mouse cytomegalovirus-encoded protein m12 [21]. The oligomeric state of Nkrp1 proteins on the surface of living cells in the presence and absence of the viral ligand remains unknown.

## 4. Materials and Methods

### 4.1. Materials

Oligonucleotide primers were obtained from KRD (Prague, CZ, Czech Republic), Generi Biotech (Hradec Králové, Czech Republic, CZ) and Sigma-Aldrich (St. Louis, MO abbreviation, USA). Superscript III reverse transcriptase and the pCR2.1-TOPO vector were purchased from Thermo Fisher Scientific (Waltham, MA, USA). The Escherichia coli BL-21 (DE3) Gold strain was from Stratagene (La Jolla, CA, USA), and the pET-30a(+) bacterial expression vectors were from Novagene (Madison, WI, USA). The pXJ41 vector was prepared by Xiao et al., 1991 [52]. Restriction endonucleases and other enzymes for DNA cloning were purchased from New England Biolabs (Ipswich, MA, USA) and Thermo Fisher Scientific (Waltham, MA, USA). The chromatographic column Superdex 75 HR 10/300 was obtained from GE Healthcare (Chicago, IL, USA). All chemicals were purchased from Sigma-Aldrich (St. Louis, MO, USA) unless otherwise declared and were of the highest available purity.

### 4.2. DNA Cloning

A template DNA for cloning manipulations was prepared from a total RNA isolated from spleens of C57BL/6 mice as described in [53]. All primers used for PCR amplification are listed in Table A2, Appendix A. DNA fragments coding mouse Nkrp1 ectodomains were amplified by RT-PCR using primers: NKRP1A STALK FW and MNKR1ARE for Nkrp1a^ECTO^ construct (residues S70-H227), NKRP1C STALK FW and MNKR1CRE to generate Nkrp1c1^ECTO^ (residues S70-S223) and Nkrp1c2^ECTO^ (S70-S220; transcriptional variant shortened by DCS sequence compared to isoform c1). Nkrp1f FW and Nkrp1 REV oligonucleotides were used for generation of Nkrp1f^ECTO^ sequence (Q67-V217). All constructs were sub-cloned into pCR2.1-TOPO vector according to the manufacturer’s protocol, amplified and then sub-cloned into pET-30a(+) bacterial expression vector between *NdeI* and *HindIII* restriction sites.

DNA encoding the full-length Nkrp1a sequence was amplified from cDNA synthetized from the total RNA using oligonucleotides Fw_Nkrp1a_Stop and Rev_Nkrp1a_Stop. Sequences of entire Nkrp1c1 and Nkrp1f were synthesized by Shanghai Generay Biotech Co., Ltd. (Shanghai, China, CN). Primers Nkrp1c1_Bam+Xho_Fw and Nkrp1c1_Bam+Xho_Rev, resp. Fw_Nkrp1f_Stop and Rev_Nkrp1f_Stop, were used to prepareNkrp1c1 gene with *BamHI* and *XhoI* restriction sites, respectively. Nkrp1f insert. DNA fragments were then sub-cloned into the *BamHI* and *XhoI* restriction sites of pXJ41 vector.

A gene encoding entire Nkrp1c2 isoform was generated by deletion of an ATTGTTCAG nucleic acid sequence of the Nkrp1c1 variant (the resulting c2 isoform lacks the DCS amino acids sequence) using primers mNKRC2Fw2 and mNKRC2Rev1. The Nkrp1c2 was ligated into vector pXJ41 between *BamHI* and *XhoI* cloning sites.

The final pXJ41-EGFP-Nkrp1 plasmids were prepared by insertion of EGFP, amplified by PCR (using primers EGFP_Eco+Bam_FW and EGFP_Eco+Bam_REV containing spacer sequence GSGGGS), between *EcoRI* and *BamHI* restriction sites on the N-terminus of the Nkrp1 sequence inserted into pXJ41 vector.

Cysteine residues in a stalk region of all Nkrp1 proteins were mutated to serines using mutation primers listed in Table A2. Wild-type sequence of the Nkrp1a in vector pXJ41 was amplified with primers A1_C220S_Fw1 and A1_C220S_Rev1, resp. A1_C262S_Fw2 and A1_C262S_Rev2 to generate pXJ41-Nkrp1a_C74S andpXJ41-Nkrp1a_C88S mutants, respectively. Double cysteine-to-serine mutant named pXJ41-Nkrp1a_C74,88S was prepared by amplification of the pXJ41-Nkrp1a_C88S utilizing primers A1_C220S_Fw1 and A1_C220S_Rev1. pXJ41-Nkrp1c1_C74,75S mutant was generated by amplification of pXJ41-Nkrp1c1 using primers C1_CC220SS_Fw and C1_CC220SS_Rev. A cysteine-to-serine mutation was introduced into pXJ41-Nkrp1f using oligonucleotides F_C220S_Fw and F_C220S_Rev to generate pXJ41-Nkrp1f_C74S version. All mutants were sub-cloned into pXJ41-EGFP plasmid between *BamHI* and *XhoI* cloning sites.

To generate a gene cassette for plasma membrane proteins with fluorescent tag in the intracellular space, EGFP constructs were prepared by PCR reaction using forward primer T198 (see Table A2) containing a *BamHI* restriction site and a spacer (GSGGGS), and reverse primer T199 containing a stop codon and an *XhoI* site. Fluorescent protein sequence was ligated into pXJ41 vector between *BamHI* and *XhoI* [54].

A pXJ41-LAT-EGFP vector was generated by Chum et al., 2016 [54]. The final construct contains sequences of 5′ UTR and leader sequence of human CD148, followed by a myc-tag and the sequence coding LAT protein followed by EGFP.

A CD8α gene was subjected to a silent mutation (residues CTG to CTC) to eliminate *EcoRI* site within the DNA construct using primers CD8a_288_Fw and CD8a_288_Rev. CD8α was then amplified by PCR utilizing oligonucleotides CD8a_Fw 1 and CD8a_REV. DNA fragment was sub-cloned into the pXJ41-EGFP vector using *EcoRI/BamHI* restriction sites to generate a pXJ41-CD8α-EGFP plasmid with intracellular fluorescent protein tag.

All cloning sequences were analyzed and confirmed by DNA sequencing.

### 4.3. Bacterial Recombinant Expression of Activating Nkrp1 Proteins

The Nkrp1a^ECTO^ (S70-H227), Nkrp1c1^ECTO^ (S70-S223), Nkrp1c2^ECTO^ (S70-S220) and Nkrp1f^ECTO^ (Q67-V217) proteins were produced using bacterial expression system following previously reported protocol [53]. Concisely, the *Escherichia coli* BL-21 (DE3) Gold cells were transformed with appropriate expression vector, the proteins produced in inclusion bodies were extracted, solubilized and refolded in vitro.

### 4.4. Protein Refolding and Purification

In vitro protein refolding was performed using modified protocol described in ref. [53]. All Nkrp1 inclusion bodies were solubilized in buffer containing 6 M guanidine-HCl (pH 8.5), 10 mM DTT and 50 mM Tris-HCl (for each 1 g of wet weight cells 8 mL of guanidine-HCl buffer was applied). Protein refolding was performed by rapid dilution method using hundred-fold higher volume of refolding buffer. The refolding buffer for Nkrp1c^ECTO^ and Nkrp1f^ECTO^ proteins consisted of 50 mM CHES (pH 9 and pH 10), 1 mM CaCl_2_, 1 M L-arginine, 100 mM NaCl, 9 mM cysteamine, 3 mM cystamine, 1 mM NaN_3_ and 1 mM PMSF. After 1–2 h of incubation at 4 °C with gentle stirring, the refolding mixtures were dialyzed twice for (4 h and 12 h) at 4 °C against 6 L of 10 mM HEPES (pH 7.4), 100 mM NaCl and 1 mM NaN_3_. Protein mixtures were then concentrated by ultrafiltration utilizing a cellulose membrane and by centrifugal filter units (MW cut-off 10 kDa, Millipore, Burlington, MA, USA).

Afterward, the Nkrp1^ECTO^ proteins were purified by size-exclusion chromatography method using a calibrated Superdex 75 column. The Nkrp1a^ECTO^ was eluted in 150 mM NaCl, 15 mM Tris-HCl (pH 7.5) and 1 mM NaN_3_. The other proteins were eluted into 10 mM HEPES (pH 7.4), 100 mM NaCl and 1 mM NaN_3_.

All protein samples were analyzed by SDS-PAGE under reducing and non-reducing conditions and the protein concentration was examined using the Bradford assay (Bio-Rad, Hercules, CA, USA) with BSA standard solution.

### 4.5. Evaluation of Disulfide Bonds

To determine disulfide bonds arrangement in recombinant Nkrp1 proteins, protein samples were subjected to non-reducing SDS-PAGE in a 4-12% polyacrylamide gradient gel with 200 µM cystamine [26,55]. The in-gel proteolytic reactions were performed using trypsin (Promega, Madison, WI, USA), Asp-N (Roche, Basel, Switzerland) and Glu-C (Roche, Basel, Switzerland) proteinases. After overnight digestion at 37 °C by 5 ng/µl of the proteinases, the digestion mixtures were desalted and analyzed by Liquid Chromatography (LC) coupled to ESI-FT-ICR MS (SolariX, Bruker Daltonics, Billerica, MA, USA). Data were interpreted utilizing software Data Analysis 4 (Bruker Daltonics, Billerica, MA, USA) and LinX.

### 4.6. Cell Culture and Transfection

COS-7 and Jurkat cells were from the cell bank of the Institute of Molecular Genetics in Prague, Czech Republic. COS-7 cells were cultured in DMEM medium (Sigma-Aldrich), Jurkat T cell line was grown in RPMI 1640 medium (Sigma-Aldrich), both supplemented with 2 mM L-glutamine and 10% fetal calf serum (both Life Technologies, Carlsbad, CA, USA). The cell lines were cultured in an incubator under controlled conditions of 37 °C, 5% CO_2_, and 95% humidity.

COS-7 cells were grown to 60–70% confluency and transiently transfected with Lipofectamine LTX Reagent (Thermo Scientific, Waltham, MA, USA) following manufacturer’s protocol. DNA-lipid complexes were formed by mixing 250 ng of DNA with 0.5 µL Lipofectamine LTX Reagent in 100 µL Opti-MEM medium (Thermo Scientific, Waltham, MA, USA) and incubated for 30 min at room temperature. COS-7 cells were incubated overnight with the mixture and imaged 18–24 h after transfection.

For western blotting, 2 million COS-7cells were seeded on 100 mm cell culture dish and transfected at >60% confluency. Cells were transfected with reaction mixture consisted of 6–8 µg of DNA, 3 mL Opti-MEM and 21 µL Lipofectamine LTX.

Jurkat cells were transiently transfected using Neon^®^ Transfection System (Life Technologies, Carlsbad, CA, USA) following modified manufacturer’s protocol, using 0.5 µg–1 µg of vector DNA per electroporation shot (3 pulses of 1400 V, each lasting 10 ms) per 200,000 cells. Further experiments were performed 18–24 h after transfection. DNA amount was varied to achieve comparable expression level for all transfected plasmids.

### 4.7. Live Cell Imaging

Cells were immobilized on poly L-lysine(PLL)-coated coverslips and imaged at 37°C in closed perfusion chamber (FCS3, Bioptech) on a commercial confocal laser scanning microscope unit (FluoView 1000, Olympus, Shinjuku, JP, Japan) equipped with 60x water immersion, NA 1.2 objective (UPlanSApo, Olympus, Shinjuku, JP, Japan). 488 nm steady-state semiconductor laser (Coherent, Santa Clara, CA, USA) was used to excite the EGFP fluorophore and combination of 560 long-pass and 505–525 band-pass filters was selected to collect the fluorescence emission.

### 4.8. Western Blotting

24 h after transfection, COS-7 cells were gently washed twice with PBS, 50 × 10^6^ cells per 1 mL of lysis buffer (20 mM Tris/HCl (pH 8.2), 100 mM NaCl, 10 mM EDTA, 1% n-decyl-β-D-maltopyranoside (Anatrace, Maumee, OH, USA), 50 mMNaF, 1 mM orthovanadate, protease inhibitor mixture (Serva, Heidelberg, DE, Germany), 40 mM iodoacetamide (Sigma Aldrich, St. Louis, MO, USA), and reducing samples containing 100 mM dithiothreitol (Serva, Heidelberg, DE, Germany) were scraped off, and incubated on ice for 30 min. The insoluble material was removed by centrifugation at 3000 rcf for 3 min at 4°C. 20 µg of the total protein was separated on SDS-PAGE using a 10% polyacrylamide gel and were transferred onto PVDF membrane (Pall Corporation, Port Washington, NYn, USA) using semi-dry blotting apparatus. The transfer was run at 0.80 mA/cm2 for up to 1.5 h. Proteins were detected with polyclonal anti-GFP antibody (Exbio, Vestec, CZ, Czech Republic) according to manufacturer’s recommendations (1:2000 dilutions). For loading control, polyclonal anti-actin antibody (Santa Cruz, Dallas, TX, USA) was used. PVDF membranes were scanned using ChemiDoc XRS+ (Bio-Rad, Hercules, CA, USA).

### 4.9. Förster Resonance Energy Transfer (Homo-FRET)

For homo-FRET analysis, transfected Jurkat cells were centrifuged for 3 min at 300 rcf at room temperature (Centrifuge 5418K, Eppendorf, Hamburg, DE, Germany), resuspended in color-free medium (RPMI Medium 1640, no phenol red, Sigma Aldrich, St. Louis, MO, USA) and landed for 10 min at 37 °C on PLL-coated 8-well chamber slides (ibidi^®^, Martinsried, DE, Germany). Cells were subsequently measured for a maximum time of 45 min under atmospheric conditions at 37 °C in the environmental chamber (OkoTouch and T Unit bold line, Okolab, Pozzuoli, IT, Italy).

Time domain anisotropy measurements were done on laser scanning confocal microscope unit (Olympus IX-71 with Microtime 200, PicoQuant, Berlin, DE, Germany) equipped with time-correlated single photon counting (TCSPC) module (PicoHarp300, PicoQuant, Berlin, DE, Germany). Time resolution of the TCSPC was set to 64 ps. 470 nm pulsed diode laser controlled by Sepia II unit (PicoQuant, Berlin, DE, Germany) was used to excite the EGFP fluorophore. Laser repetition rate was set to 20 MHz and the excitation power was modulated between 0.1–1 µW to achieve signal equivalent to 2% of the repetition frequency (i.e., 400,000 counts/s). Band pass filters (HQ525/50M, Chroma, and FF02-525/50-25, Semrock, Rochester, NY, USA) combined with long pass filter (FF01-500/LP-25, Semrock, Rochester, NY, USA) and separation dichroic mirror (Z473/635RDC, Chroma, Bellows Falls, VT, USA) selected EGFP emission. Beam-splitter polarizer cube was used to separate the parallel and perpendicular components onto separate SPAD detectors.

The G-factor correction for the instrument sensitivity ratio towards vertically and horizontally polarized light was determined using 1 µM solution of Atto 488 (Sigma-Aldrich, St. Louis, MO, USA). Emission light was collected via 60x water immersion objective (NA 1.2; UPlanSApo, Olympus, Shinjuku, JP, Japan). Low-concentration solutions of Atto 488 (Sigma-Aldrich, St. Louis, MO, USA) and purified GFP protein (Abcam, Cambridge, UK, United Kingdom) were used to determine the *L*_1_ and *L*_2_ (Equation (1)) correction factors for fluorescence depolarization caused by the high NA objective [56].

The signal from the plasma membrane was manually selected in the anisotropy images. Each of the membrane pre-selected pixels was subsequently Fourier-transformed onto phasor plot representing a map of fluorescence lifetimes. Phasor plot has been previously used in energy transfer studies [57] to distinguish fluorescent species of different lifetimes. Here, we use it only as a filter—only membrane pixels with similar fluorescence lifetime were kept for further analysis. Cells with poor phasor plots were discarded.

Subsequently, signal from all phasor-filtered pixels was summed to enhance the signal-to-noise ratio. $I_{∥}\left(t\right)$ and $I_{⊥}\left(t\right)$ curves were extracted and steady-state anisotropy (*r_steady−state_*) was calculated for each cell according to
(1)$$r_{steady-state}=\frac{\sum I_{∥}\left(t\right)-G*\sum I_{⊥}\left(t\right)}{\left(1-3*L_{2}\right)*\sum I_{∥}\left(t\right)+G*\left(2-3*L_{1}\right)*\sum I_{⊥}\left(t\right)}$$
where $I_{∥}\left(t\right)$ and $I_{⊥}\left(t\right)$ are parallel and perpendicular components of anisotropy decay, respectively, *G* is the correction factor for the instrument sensitivity towards vertically and horizontally polarized light and *L*_1_, *L*_2_ are correction factors for high NA of the objective computed from Atto 488 and GFP solutions; *L*_1_ = 0.071 and *L*_2_ = 0.015. Steady-state values obtained in one independent measurement were finally filtered and only data within two standard deviations from the mean were used for further analysis.

To visually illustrate differences between protein oligomerization states, r steady-state values were put into dot plot where each point corresponds to one cell. Additionally, two-tailed unpaired T-test was performed on whole populations to evaluate differences between individual protein variants.

## 5. Conclusions

This work presents the stoichiometry of mouse activating Nkrp1 receptors on the surface of living cells. Our results suggest that the proteins are present predominantly as dimers in the plasma membrane of mammalian cells, with the exception of the Nkrp1a isoform, which displays a more heterogeneous character. The exact positions of stalk cysteine residues influence creation of protein assembles. Further studies will be required to reveal the function of coexistence of monomeric and dimeric Nkrp1 proteins on the cell surface.

## Figures and Tables

**Figure 1 ijms-20-01884-f001:**
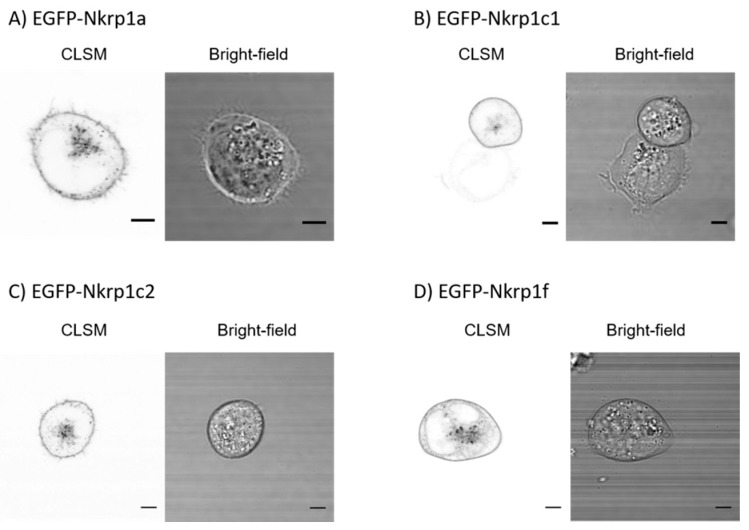
Subcellular localization of EGFP-Nkrp1 fusion proteins in living Jurkat T cells. Confocal laser scanning microscopy (CLSM) images of Jurkat cells transfected with (**A**) EGFP-Nkrp1a, (**B**) EGFP-Nkrp1c1, (**C**) EGFP-Nkrp1c2, and (**D**) EGFP-Nkrp1f. Left part of each panel shows the image from EGFP channel and right part the corresponding bright-field image. Scale bars: 5 µm.

**Figure 2 ijms-20-01884-f002:**
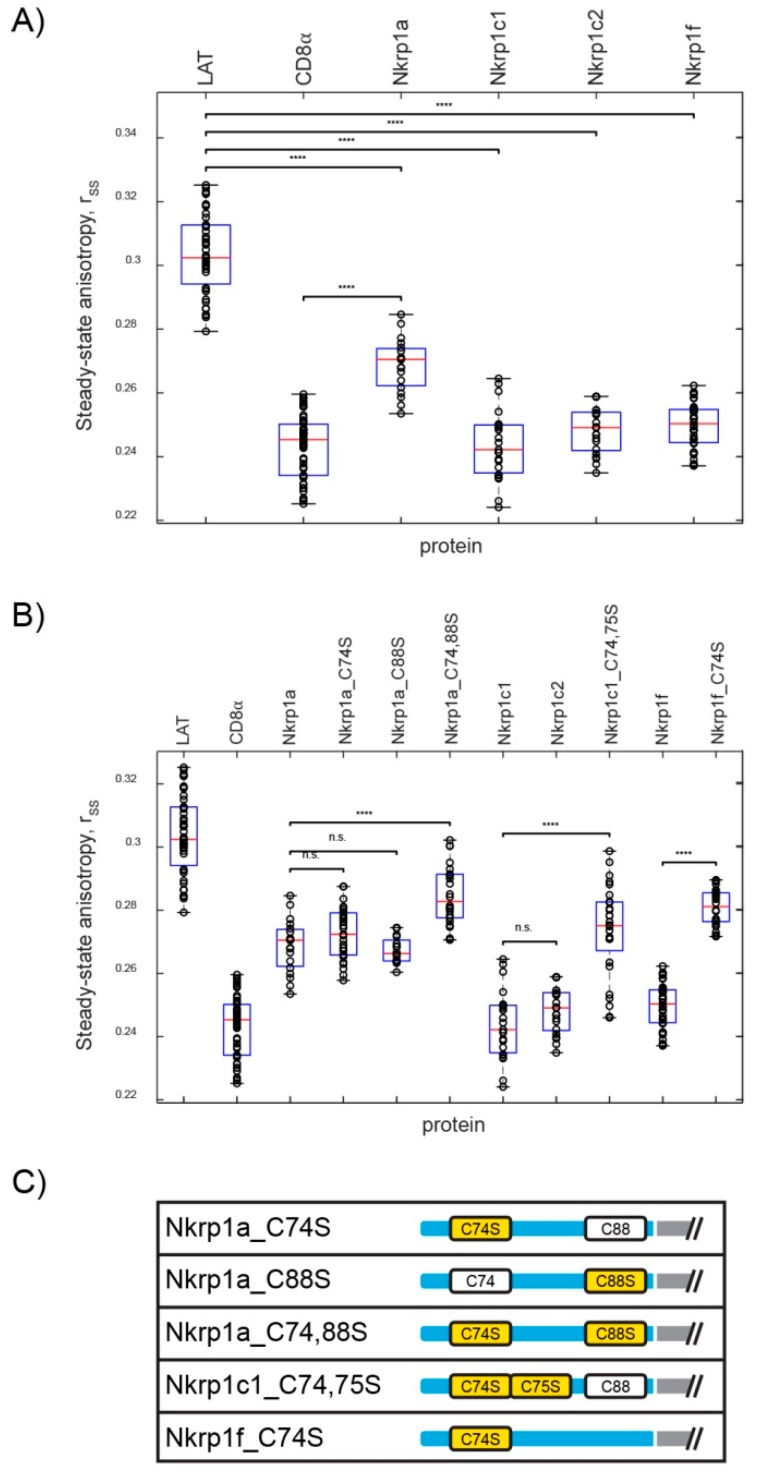
Homo-FRET measurements of EGFP-Nkrp1 proteins in living Jurkat cells. (**A**) Dot plot of steady-state anisotropy, rss, of Nkrp1a, c1, c2 and f isoforms as compared to the monomeric (LAT) and dimeric (CD8α) controls. (**B**) Assessment of the steady-state anisotropy values of mutants Nkrp1/a_C74S,/a_C88S,/a_C74,88S, Nkrp1c1_C74,75S and Nrkp1f_C74S with their native counterparts. The red line marks median of the whole set and the bottom and top edges of the box indicate the 25th and 75th percentiles, respectively. The whiskers extend to the most extreme data points not considered outliers. Statistical significance of differences is marked with asterisks: *p* < 0.00 ***, *p* < 0.0001 ****, ‘n.s.’ not significant. All median steady-state anisotropy and *p*-values are in the Appendix A
Table A2 and Table A3, respectively. (**C**) Scheme showing cysteine residues in the stalk region and their serine mutants (yellow).

**Figure 3 ijms-20-01884-f003:**
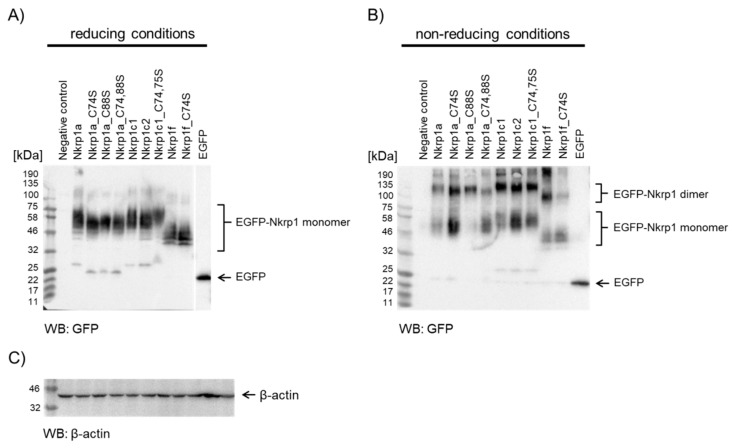
Analysis of EGFP-Nkrp1 fusion proteins by immunoblotting of reducing and non-reducing SDS-PAGE. (**A**) Immunoblotting of EGFP-Nkrp1 proteins transiently expressed in COS-7 cells, lysed and separated by SDS-PAGE under reducing conditions (100 mM DTT). All EGFP-Nkrp1 fusion proteins appear as monomers (apparent at 40-60 kDa). (**B**) Samples as in A were separated by SDS-PAGE under non-reducing conditions. EGFP-Nkrp1 fusion proteins appear as monomers, dimers and higher order oligomers with different ratio between these populations. Molecular markers are to the left of the panels. Regions corresponding to the Nkrp1 monomers and dimers, as well as EGFP only are indicated to the right. The negative control corresponds to untransfected COS-7 cells treated as all other samples. For positive control, EGFP transiently expressed in COS-7 cells was used. (**C**) Loading control: immunoblotting of β-actin detected by specific antibody at 1:1500 dilution.

**Figure 4 ijms-20-01884-f004:**
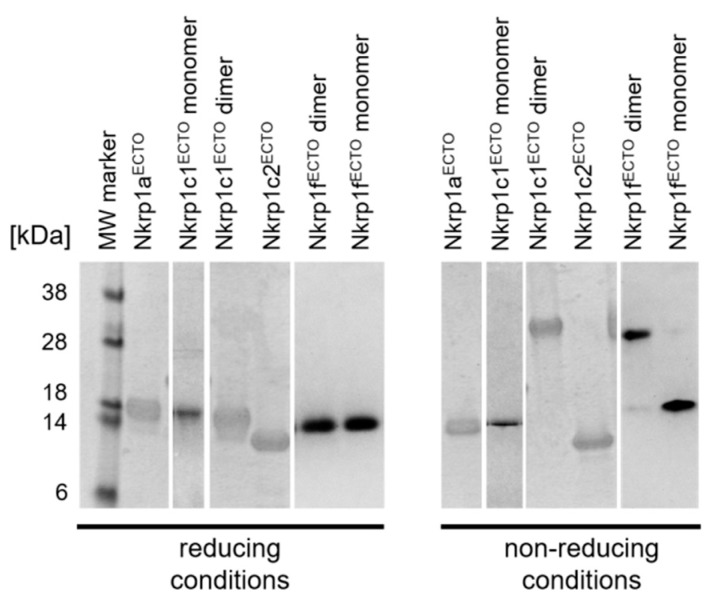
Analysis of purified recombinant Nkrp1^ECTO^ proteins. Fractions from size-exclusion chromatography of the Nkrp1^ECTO^ proteins were analyzed by 15% SDS-PAGE. Proteins appeared homogenous under reducing and non-reducing conditions. Nkrp1c1^ECTO^ and Nkrp1f^ECTO^ isoforms have formed a dimer under non-reducing conditions (≈ 35 kDa).

**Figure 5 ijms-20-01884-f005:**
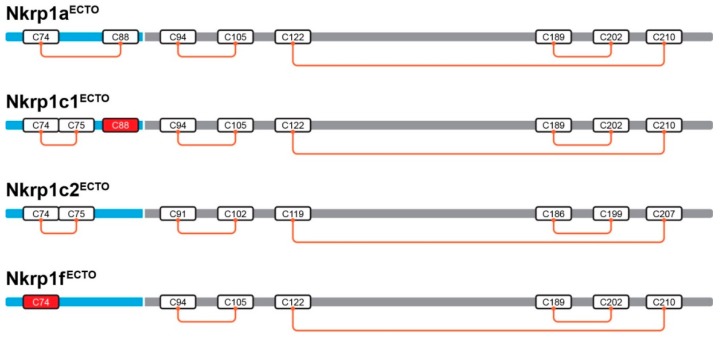
Cysteine disulfide bonds determined for the recombinant Nkrp1^ECTO^ proteins. Schematic illustration of intramolecular disulfide bonds of murine recombinant Nkrp1^ECTO^ proteins studied by LC-MS. A stalk region is indicated in blue, the extracellular C-Type Lectin-like Domain in gray (predicted by InterProScan tool) and cysteine residues involved in intermolecular bonding are in red boxes.

**Table 1 ijms-20-01884-t001:** Monoisotopic masses of intact Nkrp1^ECTO^ proteins measured by ESI-FT-ICR mass spectrometry.

Protein	Experimental [M + H]^+^, Daltons(error, ppm)	
	Native Form	Reduced Form
Nkrp1a^ECTO^	18168.5538 (0.03)	18176.6566 (2.18)
Nkrp1c1^ECTO^ monomer modified by cysteamine binding	17678.3851 (0.01)	17611.4335 (0.01)
Nkrp1c1^ECTO^ dimer	35204.4478 (7.91)	17611.5326 (5.63)
Nkrp1c2^ECTO^	17298.3286 (1.50)	17306.4254 (3.47)
Nkrp1f^ECTO^ monomer modified by cysteamine binding	17713.9178 (2.21)	17644.9799 (0.55)
Nkrp1f^ECTO^ dimer	35274.7974 (1.84)	17644.9904 (0.05)

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
