# Peer review of "Oligomeric Architecture of Mouse Activating Nkrp1 Receptors on Living Cells"

_ijms, 2019, doi:10.3390/ijms20081884_

Round 1

Reviewer 1 Report

  The authors expressed mouse activating EGFP-Nkrp1s in mammalian lymphoid cells and Forster resonance energy transfer (FRET) was used to evaluate the oligomerization. They also created several Nkrp1 protein isomers mutants to study influence of cysteine and its role.  However, limitations were present in their works:

In the “4.4 Protein refolding and purification” section (Page12, line 430), the authors introduced that refolding method was used for obtaining the target protein from inclusion body. As we know, the protein was usually problematic, and the process was difficult when refolding method was applied, especially for proteins with several cysteine residues or disulfide bridges. The details should be provided so that it will be helpful for the readers’ research work. Another related contents also should be added and uploaded, that is, how the authors verified the refolding protein were bio-active. In the following steps, interaction between protein molecules will be conducted, therefore, this step should be carefully treated without interference.  

Author Response

In the “4.4 Protein refolding and purification” section (Page12, line 430), the authors introduced that refolding method was used for obtaining the target protein from inclusion body. As we know, the protein was usually problematic, and the process was difficult when refolding method was applied, especially for proteins with several cysteine residues or disulfide bridges. The details should be provided so that it will be helpful for the readers’ research work. Another related contents also should be added and uploaded, that is, how the authors verified the refolding protein were bio-active. In the following steps, interaction between protein molecules will be conducted, therefore, this step should be carefully treated without interference. 

The reviewer is correct that there is not enough information describing the in-vitro refolding. Originally, we assumed the reference to our previous article (Rozbeský, D.; Kavan, D.; Chmelík, J.; Novák, P.; Vaněk, O.; Bezouška, K. High-level expression of soluble form of mouse natural killer cell receptor NKR-P1C(B6) in Escherichia coli. Protein Expr. Purif.2011, 77, 178–184, doi:10.1016/j.pep.2011.01.013) would be satisfactory. According to the reviewer comment we included more details in protein production session (4.4). We have no capability to provide bio-active assay on these molecules thus the success of refolding is always monitored by 1NMR spectroscopy as describe in reference 24. Since those data were already published (ref. 24) we assumed it is an appropriate to present such a data again.

Reviewer 2 Report

In the present manuscript, Adamkova et al. investigate the mechanism of dimerization of murine Nkrp1 receptors, which has been proposed to involve disulfide bonds among subunits.  The authors' goal is to determine which cysteine residues are important for homodimerization of individual Nkrp1 receptors. They use two different methods, homo-FRET and western blotting, to investigate the oligomeric composition of different Nkrp1 receptors overexpressed in mammalian cells, and investigate the identities of disulfide linkages in recombinant ectodomains of Nkrp1 receptors expressed in Escherichia coli.

In my opinion, despite the authors' attempt to provide a straightforward intepretation of the experimental data, there are significant inconsistencies between the experiments which must be clarified before reliable conclusions can be reached and the manuscript published.  

Specific comments:

1. The results obtained with homo-FRET analysis (Section 2.1.1.) do not agree with results obtained with western blotting (Section 2.1.2), despite the authors' effort to interpret them as consistent. Inconsistencies are observed both for wild-type proteins as well as mutants. In essence, the western blots show that all investigated species are present in both dimeric and monomeric forms, at ratios of roughly 1:1 (dimer vs. monomer), whereas FRET paints a picture of significant differences between individual proteins. For example, Nkrp1a is portrayed as having a stronger tendency to remain in monomeric form than other Nkrp1 proteins, which is corroborated by FRET, however, on the western blot, its behavior does not differ from other Nkrp1 proteins. The same is also true for mutant Nkrp1c1 C74,75S, etc.

2. There are inconsistencies between Figure 2 and the text in the corresponding section 2.1.1. Specifically, Figures 2B and 2C refer to mutant Nkrp1c1 C74,75S which is not mentioned in the text at all. Additionally, Figure 2C refers to mutant Nkrp1c1 C88S which is also not mentioned in the text.

3. The determined intramolecular disulfide bonds (Figure 5) in Nkrp1 proteins obtained by in vitro refolding of inclusion bodies produced in E. coli generate confusion rather than clarity. If intermolecular disulfide bonds are indeed important for the dimerization of Nkrp1 proteins, how do the authors explain dimerization of Nkrp1a and Nkrp1c2 which contain no free cysteine residues according to Figure 5? It is entirely possible that refolding produces proteins that are soluble and retain some native-like functionality but do not contain a native disulfide bond arrangement, especially in more flexible regions with less regular secondary structure. In my opinion, this could very well be the case with the stalk region of Nkrp1 proteins.

Author Response

1. The results obtained with homo-FRET analysis (Section 2.1.1.) do not agree with results obtained with western blotting (Section 2.1.2), despite the authors' effort to interpret them as consistent. Inconsistencies are observed both for wild-type proteins as well as mutants. In essence, the western blots show that all investigated species are present in both dimeric and monomeric forms, at ratios of roughly 1:1 (dimer vs. monomer), whereas FRET paints a picture of significant differences between individual proteins. For example, Nkrp1a is portrayed as having a stronger tendency to remain in monomeric form than other Nkrp1 proteins, which is corroborated by FRET, however, on the western blot, its behavior does not differ from other Nkrp1 proteins. The same is also true for mutant Nkrp1c1 C74,75S, etc.

Ad 1) The reviewer is right that live-cell homo-FRET and in vitro Western blotting (WB) data cannot be directly compared and may be interpreted as inconsistent. On the other hand, we do not see 1:1 monomer/dimer ratio for all the variants in Figure 3. We observe some differences. It is important to note that WB data suffer from the fact that incompletely processed proteins from intracellular membranes are present in post-nuclear lysates. Expression levels (poorly controlled in transient COS-7 transfectants) can influence the ratio between the surface and intracellular molecules. The sample preparation for WB involves a number of steps which may affect the stability of higher order structures. On the contrary, homo-FRET analysis focuses on the surface molecule and is concentration-independent. Proteins are in their native cell membrane environment. Therefore, our conclusions are primarily based on live-cell homo-FRET data. WB results were used more to demonstrate the existence of disulphide-linked dimers (or higher oligomers). FRET techniques cannot distinguish oligomers from densely clustered receptors.

The WB data indicate that all texted Nkrp1 isoforms tend for form dimers. As the reviewer correctly points out, Nkpr1a isoform and its mutants exhibit similar dimerisation as Nkrp1c variants in this analysis. Mutant Nkrp1a C88S was unexpectedly found almost exclusively in a dimeric form using WB method. This observation strongly contradicts FRET data. However, as mentioned, stability of proteins may influence WB data. Also, a higher presence of monomers in Nkrp1c samples may be caused by their stable presence in the intracellular membranes where receptors may be in a monomeric state. The result may be also affected by the cell type variation (which we did not observe for tested receptors using live-cell homo-FRET analysis – data not shown).

A larger proportion of monomers was detected for Nkrp1c2 than for Nkrp1c1 in lysates of COS-7 cells. Comparably high dimerization was observed in living Jurkat cells. Again, we prefer to use the WB data rather to support the existence of Nkrp1 disulfide-bond dimers and not in a fully quantitative manner. The conclusions are based primarily on the live-cell homo-FRET data.

The text of the manuscript was modified to highlight this attitude.

2. There are inconsistencies between Figure 2 and the text in the corresponding section 2.1.1. Specifically, Figures 2B and 2C refer to mutant Nkrp1c1 C74,75S which is not mentioned in the text at all. Additionally, Figure 2C refers to mutant Nkrp1c1 C88S which is also not mentioned in the text.

Ad 2) The reviewer is right. The original text combines the results of Nkrp1c1 C74,75S with those for Nkrp1f. These two mutants have to be evaluated independently. The text was revised accordingly. Moreover, Nkrp1c1 C88S mutant was added to the Figure 2C by mistake. This mutant has not been tested in this work. The scheme was modified to correct for this mistake.

 3. The determined intramolecular disulfide bonds (Figure 5) in Nkrp1 proteins obtained by in vitro refolding of inclusion bodies produced in E. coli generate confusion rather than clarity. If intermolecular disulfide bonds are indeed important for the dimerization of Nkrp1 proteins, how do the authors explain dimerization of Nkrp1a and Nkrp1c2 which contain no free cysteine residues according to Figure 5? It is entirely possible that refolding produces proteins that are soluble and retain some native-like functionality but do not contain a native disulfide bond arrangement, especially in more flexible regions with less regular secondary structure. In my opinion, this could very well be the case with the stalk region of Nkrp1 proteins.

Ad 3) The reviewer is correct and there is not a simple answer. Analysis of in-vitro refolded proteins revealed the formation intramolecular disulfide bond in the stalk region of Nkrp1a and Nkrp1c2. Sure; it can be an artifact of in-vitro refolding. On the other hand the Nkrp1a double mutant showed a little portion of oligomerization pointing out the dimerization ability of these proteins independent on disulfide formation. We preferred to present our data like that even they might raise some questions rather than saying it is a refolding artifact. If the review prefers to avoid such contradiction we agree.

Reviewer 3 Report

The manuscript titled " Oligomeric architecture of mouse activating Nkrp1  receptors on living cells" by Adámková and co-workers presents an interesting and novel approach in studding the oligomerization of Nkrp1 receptors.  Authors proved that the coexistence of monomers and dimers plays a regulatory role for the function of the aforementioned receptors and -whats more- they mapped residues that are crucial for receptor dimerization.  

Overall, the proposed work adds to the experimental study of Nkrp1 receptors and may be of interest to specialists. I only have a minor concern:

-          In most of the times, cysteine to serine mutation is a conservative change.  Authors are encouraged to comment why didn’t they introduce a cysteine to alanine substitution and maybe -computationally- prove the advantages of a serine instead of serine substitution in their assays. A computational alanine scanning may add to the experimental results presented in this work.

Author Response

The manuscript titled " Oligomeric architecture of mouse activating Nkrp1  receptors on living cells"  by Adámková and co-workers presents an interesting and novel approach  in studding the oligomerization of Nkrp1 receptors.  Authors proved that  the coexistence of monomers and dimers plays a regulatory role for the  function of the aforementioned receptors and -whats more- they mapped  residues that are crucial for receptor dimerization.  

Overall,  the proposed work adds to the experimental study of Nkrp1 receptors and  may be of interest to specialists. I only have a minor concern:

-          In  most of the times, cysteine to serine mutation is a conservative  change.  Authors are encouraged to comment why didn’t they introduce a  cysteine to alanine substitution and maybe -computationally- prove the  advantages of a serine instead of serine substitution in their assays. A  computational alanine scanning may add to the experimental results  presented in this work.

Yes, it is a nice comment. We introduced the Cys to Ser substitutions because Ser and Cys differ only in the swap of a sulfur atom with oxygen and share a high level of similarity in terms of size and structure. Furthermore, all mutations were located in the stalk region which we expect to be fully exposed to the solvent. Thus, the substitution to Ser residue rather than Ala residue, which is more hydrophobic, appears to be more meaningful. According to this comment we included a text to the paragraph 2.1. Unfortunately, we were not able to computationally show the advantages of the Cys to Ser instead of Ala to Ser substitutions because structures of the full-length ectodomains of Nkrp1 receptors have not been reported to date. Previous studies revealed structures of the CTLD domains without the stalk region. Also, computational analysis of stalk regions is usually hampered by their higher flexibility relative to the well-ordered domains.

Round 2

Reviewer 2 Report

The authors made an effort to reply to my comments and some modifications to the manuscript. Though I am still not  convinced by the presentation of data regarding refolded proteins, I agree that these data can be interpreted in different ways. I think the manuscript's present state is acceptable.